# Electricity Consumption Forecasting using Support Vector Regression with the Mixture Maximum Correntropy Criterion

**DOI:** 10.3390/e21070707

**Published:** 2019-07-19

**Authors:** Jiandong Duan, Xuan Tian, Wentao Ma, Xinyu Qiu, Peng Wang, Lin An

**Affiliations:** 1School of Automation and Information Engineering, Xi’an University of Technology, Xi’an 710048, China; 2State Key Laboratory of Electrical Insulation and Power Equipment, Xi’an Jiaotong University, Xi’an 710048, China; 3School of Statistics, Xi’an University of Finance and Economics, Xi’an 710100, China

**Keywords:** electricity consumption forecasting, support vector regression, mixture maximum correntropy criterion, parameter optimization

## Abstract

The electricity consumption forecasting (ECF) technology plays a crucial role in the electricity market. The support vector regression (SVR) is a nonlinear prediction model that can be used for ECF. The electricity consumption (EC) data are usually nonlinear and non-Gaussian and present outliers. The traditional SVR with the mean-square error (MSE), however, is insensitive to outliers and cannot correctly represent the statistical information of errors in non-Gaussian situations. To address this problem, a novel robust forecasting method is developed in this work by using the mixture maximum correntropy criterion (MMCC). The MMCC, as a novel cost function of information theoretic, can be used to solve non-Gaussian signal processing; therefore, in the original SVR, the MSE is replaced by the MMCC to develop a novel robust SVR method (called MMCCSVR) for ECF. Besides, the factors influencing users’ EC are investigated by a data statistical analysis method. We find that the historical temperature and historical EC are the main factors affecting future EC, and thus these two factors are used as the input in the proposed model. Finally, real EC data from a shopping mall in Guangzhou, China, are utilized to test the proposed ECF method. The forecasting results show that the proposed ECF method can effectively improve the accuracy of ECF compared with the traditional SVR and other forecasting algorithms.

## 1. Introduction

### 1.1. Motivation

Electricity consumption forecasting (ECF) refers to the estimation or forecast of the future monthly electricity sales by collating and analyzing historical data. Whether for the bid evaluation system of the State Grid Corporation or for the open electricity market today, electricity sales forecasting is very important, especially for power sales companies, since it will directly affect the deviation assessment. The bigger the deviation, the higher the penalty. Accurate prediction of monthly electricity sales is of vital practical significance and practical value to the power sales company.

ECF is a new problem challenging the current power sales company. In essence it consists of load forecasting. However, compared with load forecasting, it has some new characteristics, such as fewer historical load data and stronger randomness. 

### 1.2. Literature Review

Extensive research has been conducted on load forecasting. Ref. [1] established a long-term power load forecasting model by using a support vector machine (SVM) model based on the comprehensive consideration of economic factors, social factors, and energy market structure and optimizing a multi-factor medium and the parameters of SVM by using particle swarm optimization (PSO) and an improved cross-validation method. In Ref. [2], the improved fly optimization algorithm (FOA) algorithm was used to optimize the parameters of the wavelet least-squares support vector machine (using the wavelet kernel function instead of the Gauss kernel function in least-squares support vector machines (LSSVM)), and the combined algorithm was applied to medium- and long-term power load forecasting. In Ref. [3], the author combined the Fuzzy Time Series with Seasonal Autoregressive Fractionally Integrated Moving model to predict load changes, and the PSO algorithm s used to optimize the weights of the model leading to a good prediction result. In Ref. [4], the author proposed a hybrid optimization grey model called Rolling-ALO-GM (1,1) for ECM based on known information from annual EC by using the grey control theory. In Ref. [5], the author used the Drosophila optimization algorithm with decreasing step size (SFOA) to optimize the extended parameters of generalized regression neural network (GRNN). By combining it with the periodic characteristics of weather factors and power load, a SFOA–GRNN power load forecasting model was proposed. The experimental results showed that the improved model was much better than the original model. 

The SVM is a supervised learning algorithm which can solve the problems of small sample, non-linearity, and high latitude and is more efficient in solving pattern recognition and regression problems. Therefore, it has been widely used in short-term load forecasting. In Ref. [6], in order to improve the accuracy and efficiency of calculation, the secondary sampling strategy of support vector regression (SVR) learning process was designed, and a method of modeling and forecasting short-term load point estimation and its confidence interval length using subsampled SVR ensemble was obtained. In Ref. [7], a new grid algorithm was proposed, which provides a new method for the determination of parametric regression surface. A sequential grid method based on SVR was established for short-term load forecasting. Compared with the standard SVR model, the accuracy of short-term load forecasting was greatly improved. In order to deal with the cyclic nature of electric loads, Ref. [8] proposed the SVR with chaotic cuckoo search (SVRCCS) model, based on using a tent chaotic mapping function to enrich the cuckoo search space and diversify the population to avoid trapping in local optima, and combined the SVRCCS model with a seasonal mechanism, thus obtaining a seasonal SVRCCS model. The numerical results showed that the proposed novel model outperforms other alternative models. Ref. [9] used the proposed chaotic eﬃcient bat algorithm, niche search, and evolution mechanisms to optimize the parameters of the hybrid kernel-based SVR model and used motion data for a real floating platform to evaluate the reliability and effectiveness of the proposed model. Ref. [10] proposed a novel vector field-based SVR method. Through multi-distortions in the sample data space or high-dimensional feature space mapped by a vector field, the optimal feature space was found, in which the high non-linearity between inputs and outputs was approximated by linearity. The results indicated that the proposed method achieves better performance than commonly used methods with regard to the accuracy, robustness, and generalization ability.

However, the loss function of the traditional SVR is the mean-square error (MSE) criterion. The MSE criterion only considers the second-order distance of the prediction error distribution. It is very effective for data with Gaussian characteristics, which results in high accuracy only when predicting a stationary sequence. Compared with the power consumption of large users and stable users, the power consumption of small users is constrained by many external factors, such as orders, operation plans, equipment, environmental factors, meteorological changes, emergencies, and so on. It produces non-linear characteristics, such as randomness and uncertainty, and prediction errors are often non-gaussian. Therefore, the application of MSE-based SVR in power consumption forecasting will be less accurate compared with that for large users and stable users. It produces non-linear characteristics, such as randomness and uncertainty, and prediction errors are often non-gaussian. Therefore, the application of SVR based on MSE in ECF has some shortcomings. The Neural Network Laboratory of the University of Florida, USA, proposed a new definition of the correlation function of stochastic processes, namely, correntropy [11]. When it is taken as the cost function in machine learning and signal processing, a Maximum Correntropy Criterion (MCC) can be defined for non-gauss, non-linear applications. The MCC contains higher moments of probability density function and is suitable for any noise environment. So far, many robust learning algorithms have been developed using MCC [12,13,14,15,16,17,18]. It has been proved that they can perform a robust analysis [19,20,21,22] and effectively deal with non-Gaussian situations and outliers [23,24,25]. In order to further improve the learning performance, Ref. [26] proposed the concept of the maximum mixture correntropy criterion (MMCC), which uses the mixture of two Gauss functions as the core function. The learning algorithm using MMCC has a better performance than the traditional MCC-based learning algorithm and several other advanced learning algorithms. On the basis of the above analysis, MMCC was introduced into SVR for the first time, and an MMCCSVR forecasting model for EC is proposed in this paper.

### 1.3. Our Contribution

The main contributions of this study can be summarized as follows:

(1) In order to solve the problem due to the fact that traditional SVR based on MSE loss function only has high efficiency in data processing with Gaussian characteristics, which leads to its insensitivity to burst and low accuracy in predicting a non-stationary sequence, MMCC is introduced into SVR as loss function, and an MMCCSVR model is proposed.

(2) The MMCCSVR model is applied to ECF for the first time. The simulation results show that the forecasting accuracy of this method is higher than that of traditional SVR and other algorithms.

### 1.4. Organization of the Paper

This paper is structured as follows: Section 2 gives an overview of SVR and MMCC. Section 3 proposes the MMCCSVR model for ECF in detail. Section 4 introduces the main steps of forecasting. In Section 5, the model validation is introduced and next, experimental results and comparisons with traditional SVR and other algorithms are shown and discussed. Finally, a brief summary and a description of future work are given in Section 6.

## 2. Methodology 

### 2.1. SVR

The SVR model has its own unique advantages in dealing with such problems as small amount of data, obvious non-linear characteristics, and high-dimensional pattern recognition. It only needs a small number of support vectors to establish decision functions, which can effectively avoid dimension disasters. Moreover, the reduction or increase of non-support vector data has little impact on the algorithm and has better robustness. SVR uses the kernel function φ(x) in substitution of nonlinear mapping, the controlled parameters are few, and the algorithm is simple and easy to implement [27].

In an SVR model, suppose the training set sample is $\left(x_{i},y_{i}\right)|i=1,2,\ldots ,n$, and n is the number of samples. The regression problem can be transformed into an optimization problem, as follows.
(1)$$\begin{array}{l} \min\;C\sum_{i=1}^{n}\xi_{i}^{2}+{\left\|\omega \right\|}^{2}\\ s.t.\;\omega^{T}\varphi \left(x_{i}\right)+b-y_{i}=\xi_{i},i=1,2,\ldots ,n \end{array}$$
where ξ is the error variable, ω is the weight vector, ω∈H, b is the bias, and C is the penalty parameter; C is used to control the minimization of the estimation error and the function smoothness.

In order to solve the optimization problem, the Lagrange function is developed as
(2)$$L\left(\omega ,b,\xi ,\alpha \right)=\left(\omega^{T}\omega \right)+C\sum_{i=1}^{n}\xi_{i}^{2}-\sum_{i=1}^{n}\alpha_{i}^{2}\left[\omega^{T}\varphi \left(x_{i}\right)+b-y_{i}-\xi_{i}\right]$$
where $\alpha =\left(\alpha_{1},\alpha_{2},\ldots ,\alpha_{n}\right)$ is the Lagrange multiplier. Differentiating *L* with respect to the variables ω,b,ξ, and α, we obtain
(3)$$\frac{\partial L}{\partial \omega }=0\to \omega =\sum_{I=1}^{n}\alpha_{i}\varphi \left(x_{i}\right),$$
(4)$$\frac{\partial L}{\partial b}=0\to \omega =\sum_{I=1}^{n}\alpha_{i}=0,$$
(5)$$\frac{\partial L}{\partial \xi_{i}}=0\to \alpha_{i}=C\xi_{i},$$
(6)$$\frac{\partial L}{\partial \alpha_{i}}=0\to \omega^{T}\varphi \left(x_{i}\right)+b+\xi_{i}-y_{i}=0,$$
and, after solving the above functions, we can obtain the solution of the problem in the following form
(7)$$f\left(x\right)=\sum_{i=1}^{n}\omega_{i}K\left(x,x_{i}\right)+b$$
where K(⋅) is the kernel function. In this paper, we used Gaussian kernel $K\left(x,x_{i}\right)=\exp\left[-\frac{{\left\|x-x_{i}\right\|}^{2}}{\gamma^{2}}\right]$ to construct nonlinear SVR since it is widely used in the literature and has shown good learning properties in a variety of applications.

In the following subsection, we introduce the MMCC method.

### 2.2. Mixture Maximum Correntropy Criterion

First, correntropy is defined as follows
(8)V(X,Y)=E[κ(X,Y)]
where κ(X,Y) is the kernel function, and E[•] is the mathematical expectation. Correntropy is a generalized correlation measure. Suppose ${\left\{\left(x_{i},y_{i}\right)\right\}}_{i=1}^{n}$ is of the random variables X, Y, the sample mean estimator of corrrentropy is
(9)$$\hat{V}\left(X,Y\right)=\frac{1}{N}\sum_{i=1}^{N}g\left(x_{i}-y_{i},\sigma \right)$$
where $g\left(x_{i}-y_{i},\sigma \right)=G_{\sigma }\left(•\right)=\exp\left(-\frac{{\left\|x_{i}-y_{i}\right\|}^{2}}{2\sigma^{2}}\right)$ is the Gaussian kernel with bandwidth σ>0. 

As a nonlinear similarity measure, correntropy has been successfully used as an efficient optimization cost in signal processing and machine learning [25], and the corresponding cost function is MCC. In order to further improve the learning performance, Ref. [25] proposed a mixture correntropy that uses the mixture of two Gaussian functions as the kernel function; the mixture correntropy is defined by
(10)$$M\left(X,Y\right)=E\left[\lambda G_{\sigma_{1}}\left(X-Y\right)+\left(1-\lambda \right)G_{\sigma_{2}}\left(X-Y\right)\right]$$
where $\sigma_{1}$ and $\sigma_{2}$ are the kernel bandwidths of the Gaussian functions $G_{\sigma_{1}}\left(•\right)$ and $G_{\sigma 2}\left(•\right)$, and 0≤λ≤1 is the mixture coefficient. Hence, Equation (10) can be written as
(11)$$\hat{M}\left(X,Y\right)=\frac{1}{n}\sum_{i-1}^{n}\left(\lambda g\left(x_{i}-y_{i},\sigma_{1}\right)+\left(1-\lambda \right)g\left(x_{i}-y_{i},\sigma_{2}\right)\right)$$
Especially, when $\sigma_{1}=\sigma_{2}$, MMCC degenerates into MCC. 

## 3. SVR with MMCC 

The central idea of SVR with MMCC (MMCCSVR) is to integrate the MMCC regularization technique and the kernel trick in a unified framework. Therefore, the MMCCSVR seeks the optimal regression function $f\left(x\right)=\sum_{i=1}^{n}\omega_{i}K\left(x,x_{i}\right)+b$ by solving the following optimization problem
(12)$$\underset{\omega ,b}{\max}\;J\left(\omega ,b\right)=C\sum_{i=1}^{n}\left(\lambda g\left(\omega^{T}\varphi \left(x_{i}\right)+b-y_{i},\sigma_{1}\right)+\left(1-\lambda \right)g\left(\omega^{T}\varphi \left(x_{i}\right)+b-y_{i},\sigma_{2}\right)\right)-{\left\|\omega \right\|}^{2}$$

Concretely, the first term of (12) is based on the aforementioned MMCC (11) in the φ−included kernel space and is used as an empirical loss function to measure the fitting error. The second term of (12) makes the required regression estimation f(x) as smooth as possible from the perspective of Tikhonov regularization. It can also be interpreted as a regularization to overcome possible overfitting. 

However, there are two difficulties to solve (12) directly. The first one is that the correntropy cost (11) is hard to be optimized. Although Liu et al. proposed a steepest descend-based method in Ref. [27] to tackle this problem, it is not very efficient. The second problem concerning (12) is due to the fact that ω may lie in a high or even infinite dimensional feature space induced by nonlinear map φ(⋅), thus limiting the application of conventional optimization methods. Fortunately, (12) can be converted to a more straightforward form by considering the following proposition to tackle the first problem [28,29,30].

**Proposition** **1.***There exists a convex function*ϕ: R→R, *such that*(13)$$g\left(x,\sigma \right)=\underset{p<0}{\max}\left(p\frac{{\left\|x\right\|}^{2}}{\sigma^{2}}-\phi \left(p\right)\right)$$*and, for a fixed*x, *the maximum is reached at*p=−g(x,σ). 

The proof of Proposition 1 is based on the convex conjugated function theory and can be found in Ref. [31]. Now, we construct the following optimization problem with augmented objective function in an enlarged parameter space:(14)$$\underset{\omega ,b,p}{\max}\bar{J}\left(\omega^{T},b,p\right)=C\sum_{i=1}^{n}\left(\lambda p_{i}\frac{{\left(\omega^{T}\varphi \left(x\right)+b-y_{i}\right)}^{2}}{\sigma_{1}{}^{2}}+(1-\lambda )p_{i}\frac{{\left(\omega^{T}\varphi \left(x\right)+b-y_{i}\right)}^{2}}{\sigma_{2}{}^{2}}-\phi \left(p_{i}\right)\right)-{\left\|\omega \right\|}^{2}$$
where $p={\left[p_{1},p_{2},\ldots ,p_{n}\right]}^{T}$ stores the auxiliary variables introduced in the half-quadratic optimization. Then, according to proposition 1, we notice that when [ω,b] is fixed, the following equation holds
(15)$$J\left(\omega ,b\right)=\underset{p}{\max}\bar{J}\left(\omega ,b,p\right)$$
Hence, we can further get the following equation
(16)$$\underset{\omega ,b}{\max}J\left(\omega ,b\right)=\underset{\omega ,b,p}{\max}\bar{J}\left(\omega ,b,p\right)$$

That is to say, maximizing J(ω,b) with respect to [ω,b] is equivalent to maximizing the augmented function $\bar{J}\left(\omega ,b,p\right)$ in the enlarged parameter space [ω,b,p]. Now, to overcome the second problem, we introduce a slack vector to convert the unconstrained optimization problem (16) to the following constrained problem $\xi ={\left[\xi_{1},\xi_{2},\ldots ,\xi_{n}\right]}^{T}$, as in standard SVR
(17)$$\begin{array}{ll} \underset{\omega ,b,\xi ,p}{\max} & \frac{C}{2}\sum_{i=1}^{n}\left(\lambda p_{i}\frac{\xi_{i}^{2}}{\sigma_{1}{}^{2}}+(1-\lambda )p_{i}\frac{\xi_{i}^{2}}{\sigma_{2}{}^{2}}-\phi \left(p_{i}\right)\right)-\frac{1}{2}{\left\|\omega \right\|}^{2}\\ s.t. & \omega^{T}\varphi \left(x_{i}\right)+b-y_{i}=\xi_{i},i=1,2,\ldots ,n \end{array}$$

We can choose to iteratively optimize (17) by alternating the optimization with respect to either $\left[\omega ,b,\xi_{i}\right]$ or *p*, while holding the other term fixed. 

First, we hold *p* fixed and maximize (17) with respect to $\left[\omega ,b,\xi_{i}\right]$. By dropping the other unrelated variables and introducing the variable $q_{i}=-p_{i}$, we obtain the following equivalent problem
(18)$$\begin{array}{ll} \underset{\omega ,b,\xi }{\min} & \frac{C}{2}\sum_{i=1}^{n}\left(\lambda q_{i}\frac{\xi_{i}^{2}}{\sigma_{1}{}^{2}}+(1-\lambda )q_{i}\frac{\xi_{i}^{2}}{\sigma_{2}{}^{2}}\right)+\frac{1}{2}{\left\|\omega \right\|}^{2}\\ s.t. & \omega^{T}\varphi \left(x_{i}\right)+b-y_{i}=\xi_{i},i=1,2,\ldots ,n \end{array}$$

In order to solve the above problem, we introduce the following Lagrangian function
(19)$$L=C\sum_{i=1}^{n}\left(\lambda q_{i}\frac{\xi_{i}^{2}}{2\sigma_{1}{}^{2}}+\left(1-\lambda \right)q_{i}\frac{\xi_{i}^{2}}{2\sigma_{2}{}^{2}}\right)+\frac{1}{2}{\left\|\omega \right\|}^{2}-\sum_{i=1}^{m}\alpha_{i}(\omega^{T}\varphi \left(x_{i}\right)+b-y_{i}-\xi_{i})$$
where $\alpha_{i}$ is a Lagrangian multiplier. Then, by following the Karush-Kuhn-Tucker (KKT) condition [32], we have
(20)$$\frac{\partial L}{\partial \omega }=\omega -\sum_{i=1}^{n}\alpha_{i}\varphi \left(x_{i}\right)=0\to \omega =\sum_{i=1}^{n}\alpha_{i}\varphi \left(x_{i}\right)$$
(21)$$\frac{\partial L}{\partial b}=0\to \sum_{i=1}^{n}\alpha_{i}=0$$
(22)$$\frac{\partial L}{\partial \xi_{i}}=\frac{Cq_{i}\lambda \xi_{i}}{\sigma_{1}{}^{2}}+\frac{Cq_{i}\left(1-\lambda \right)\xi_{i}}{\sigma_{2}{}^{2}}+\alpha_{i}=0\to \xi_{i}=-\frac{\alpha_{i}\sigma_{1}^{2}\sigma_{2}^{2}}{\lambda Cq_{i}\sigma_{2}^{2}+\left(1-\lambda \right)Cq_{i}\sigma_{1}^{2}}$$
(23)$$\frac{\partial L}{\partial \alpha_{i}}=0\to \omega^{T}\varphi \left(x_{i}\right)+b-y_{i}=\xi_{i}$$

Combing Equations (20)–(23) leads to the following linear system of equations
(24)$$\left[\begin{array}{ll} K+Q & e\\ e^{T} & 0 \end{array}\right]\left[\begin{array}{l} \alpha \\ b \end{array}\right]=\left[\begin{array}{c} Y\\ 0 \end{array}\right]$$
where K is an n×n kernel matrix with $K_{ij}=\phi {\left(x_{i}\right)}^{T}\phi {\left(x_{j}\right)}^{T}=K\left(x_{i},x_{j}\right)$, $Y={\left[y_{1},y_{2},\ldots ,y_{n}\right]}^{T}$, $e={\left[1,1,\ldots ,1\right]}^{T}$, and Q is a diagonal matrix whose diagonal element $Q_{ii}=\left(\frac{\sigma_{1}^{2}\sigma_{2}^{2}}{\lambda Cq_{i}\sigma_{2}^{2}+\left(1-\lambda \right)Cq_{i}\sigma_{1}^{2}}\right)=-\left(\frac{\sigma_{1}^{2}\sigma_{2}^{2}}{\lambda Cp_{i}\sigma_{2}^{2}+\left(1-\lambda \right)Cp_{i}\sigma_{1}^{2}}\right)>0$, since $p_{i}<0$. Noticing that K+Q is symmetric positive-definite and thus invertible, we can efficiently solve (23) by the following equations [33].
(25)$$b=\frac{e^{T}{\left(K+Q\right)}^{-1}Y}{e^{T}{\left(K+Q\right)}^{-1}e}\;\mathrm{and}\;\alpha ={\left(K+Q\right)}^{-1}\left(Y-eb\right)$$

Second, we hold [ω,b,ξ] fixed and optimize (17) with respect to *p*. Actually, according to Proposition 1, the optimal *p* is directly given by
(26)$$p_{i}=-g\left(\xi_{i},\sigma \right)=-\exp\left(-\frac{\lambda \xi_{i}^{2}}{\sigma_{1}{}^{2}}+\left(1-\lambda \right)\frac{\xi_{i}^{2}}{\sigma_{2}{}^{2}}\right)$$

Up to now, we have solved both sub-problems constituting the original optimization problem (17).

For the sake of clarity, we prescribe the proposed MMCCSVR as follows.

First, set the parameter values $p_{i}=-q_{i}=-1$ for all samples, tolerance ε=1e−3, MMCC parameter $\sigma_{1}$ and $\sigma_{2}$, kernel parameter γ, regularization parameter C, build kernel matrix K.

**Step 1**. Calculate the diagonal matrix Q and then solve (25) by Cholesky factorization to obtain [α,b].

**Step 2**. If [α,b] changes less than ε, go to Step 4, otherwise go to Step 3.

**Step 3**. Calculate the error variable $\xi_{i}$ by (22) and update $p_{i}=-q_{i}$ by (26) and then go to Step 1.

**Step 4**. Determine the final regression estimation by (7).

Referring to the convergence analysis method in Ref. [28,29,30], the convergence of the proposed algorithm is proved as follows.

According to Step 1 and Step 2, we have $\bar{J}\left(\omega^{t},b^{t},p^{t}\right)\leq \bar{J}\left(\omega^{t+1},b^{t+1},p^{t}\right)$ and $\bar{J}\left(\omega^{t+1},b^{t+1},p^{t}\right)\leq \bar{J}\left(\omega^{t+1},b^{t+1},p^{t+1}\right)$. Hence, we can conclude that the sequence $\bar{J}\left(\omega^{t},b^{t},p^{t}\right)$, t=1,2,… is non-decreasing. Based on the property of correntropy, we can verify that the objective function $\bar{J}\left(\omega ,b,p\right)$ is bounded above, since g(x,σ) and $-{\left\|\omega \right\|}^{2}$ are both bounded above. Due to the above facts and the well-known monotone convergence theorem, the algorithm is guaranteed to converge to its local optimal solution.

## 4. Electricity Consumption Forecasting Based on MMCCSVR

### 4.1. Characteristic Analysis of Electricity Consumption Data

By analyzing the characteristics of users’ EC, power sales company can accurately understand the demand response mechanism and formulate scientific marketing strategies. This is of great significance for peak shaving and valley filling, optimizing the EC curve, and improving power quality.

The monthly EC of a shopping mall in Guangdong Province in 2017 was analyzed as an example. Figure 1 shows the daily EC of the commercial property common area of the mall and its corresponding daily maximum temperature curve for 2017, which are normalized values. As shown in Figure 1, the greatest influence factor of Heating, Ventilation and Air Conditioning (HVAC) EC is the temperature, and it can be seen that the total EC of the property public area of the shopping mall fluctuated, obviously, according to the season and had strong non-linear and fluctuating characteristics. Besides the similar fluctuating trends of HVAC and the total EC of the property public area of the shopping mall, the other sub-items showed a steady trend and did not change with time.

Therefore, considering the influence of HVAC EC, it was possible to forecast the EC of the whole commercial property public area and further analyze the influencing factors of HVAC EC.

Comparing the trends of daily air conditioning power consumption and daily maximum temperature, we obtained Figure 2. 

In order to further study their correlation, we calculated Pearson correlation coefficients between historical HVAC daily EC values and their corresponding daily maximum temperature values.
(27)$$\begin{array}{ll} \rho_{X,Y} & =\frac{\mathrm{cov}(X,Y)}{\delta_{X},\delta_{Y}}\\ & =\frac{E\left[\left(X-\mu_{X}\right)\left(Y-\mu_{Y}\right)\right]}{\delta_{X},\delta_{Y}}\\ & =0.9214 \end{array}$$

The results showed that there was a great positive correlation between HVAC EC and air temperature. Therefore, air temperature and historical EC were input in the prediction model in order to improve the prediction accuracy of user EC.

### 4.2. Data Preprocessing

In the process of forecasting, the difference of magnitude of the data may affect the accuracy of forecasting and the speed of forecasting, so the initial data should be normalized to ensure the accuracy of the forecasting results. In the equation below, $y_{i}{}^{\prime }$ indicates the normalized data. The formula for data normalization is as follows:(28)$$y_{i}{}^{\prime }=\frac{y_{i}-y_{\min}}{y_{\max}-y_{\min}}$$

Generally, the initial data are normalized to [0,1] to ensure the speed and accuracy of the forecasting process.

After the prediction results are obtained, the obtained results should be anti-normalized to get the actual value of the predicted electricity. The inverse normalization formula is as follows:(29)$$y_{i}=\left(y_{\max}-y_{\min}\right)\times y_{i}^{\prime }+y_{\min}$$

### 4.3. Parameter Optimization

In order to ensure the accuracy of the results and the stability of the model, it is necessary to optimize the parameters of the ECF model. In this paper, the grid optimization method and K-fold cross-validation were combined to optimize the parameters.

It is convenient and simple to use the grid optimization method to optimize the parameters, because it is faster than other intelligent algorithms and can generally find a global optimal solution. The algorithm represents each set of parameters in the form of a range grid, traverses each set of parameters by using the independence of each grid, and then verifies the rationality of the set of parameters by K-fold cross-validation theory. Based on experience, most of the parameters $\sigma_{1}$ and $\sigma_{2}$ of MMCC were selected in [0, 50]. In the SVR model, both C and the kernel parameter γ have a wide range of values, but a too large range of values will lead to a lower search speed. After many experiments, the probability that γ falls into the [0, 1] interval is higher. Therefore, the grid range of the model parameters were specified in this paper as C=[0,20], γ=[0,1], $\sigma_{1}=[0,50]$, $\sigma_{2}=[0,50]$. 

### 4.4. Model Implementation 

This paper proposes an ECF model based on MMCCSVR considering the users’ characteristics. Based on the robustness of MMCC in non-Gaussian sequence processing, the MMCCSVR ECF model was established to predict future EC. The parameters were optimized by combining grid optimization and K-fold cross-validation. The prediction model is
(30)$$f\left(x\right)=\sum_{i=1}^{n}\omega_{i}K\left(x,x_{i}\right)+b$$
where x is input data, including historical EC and corresponding temperature data, f(x) is the forecasting results of EC.

The specific implementation process is shown in Figure 3. 

### 4.5. Evaluation Criterion

In order to comprehensively and extensively evaluate the effectiveness of the proposed forecasting model, five widely used performance metrics in statistics, i.e., mean absolute error (MAE), root-mean-square error (RMSE), mean absolute percentage error (MAPE), and coefficient of determination (R^2^), were considered. The MAE, RMSE, and MAPE were used to measure the deviation between the actual EC and the prediction load. In addition, R^2^ indicates the extent to which the EC is predictable. These five metrics are defined as follows:(31)$$MAE=\frac{1}{N}\sum_{i=1}^{N}\left|f\left(x\right)-y_{i}\right|\times 100\%$$
(32)$$MAPE=\frac{1}{N}\sum_{i=1}^{N}\frac{\left|f\left(x\right)-y_{i}\right|}{y_{i}}\times 100\%$$
(33)$$RMSE=\sqrt{\frac{1}{N}\sum_{i=1}^{N}{\left(f\left(x\right)-y_{i}\right)}^{2}}\times 100\%$$
(34)$$R^{2}=\frac{{\left(N\sum_{i=1}^{N}y_{i}\times f\left(x\right)-\sum_{i=1}^{N}y_{i}\sum_{i=1}^{N}f\left(x\right)\right)}^{2}}{\left[N\sum_{i=1}^{N}f{\left(x\right)}^{2}-{\left(\sum_{i=1}^{N}f\left(x\right)\right)}^{2}\right]\cdot \left[N\sum_{i=1}^{N}y_{i}{}^{2}-{\left(\sum_{i=1}^{N}y_{i}\right)}^{2}\right]}$$

## 5. Results

In order to verify the practicability and forecasting efficiency of the proposed ECF method, we considered the EC data of a shopping mall in Guangdong Province from 1 January 2018, to 3 June 2018, and the corresponding daily maximum temperature as the research object. We selected the daily EC from 1 January to 3 May as the training sample, forecasted the EC of 31 days from 4 May to 3 June 2018, one day ahead, and evaluated comprehensively the ECF model based on MMCCSVR.

### 5.1. Parameters Selection 

In this subsection, the joint efficiency of the free parameters for the prediction accuracy was investigated by using the grid optimization and conducting many experiments. Because the penalty coefficient C showed various empirical values in different applications, and different λ values correspond to a set of different optimal $\sigma_{1}$ and $\sigma_{2}$, which can make the prediction accuracy meet the requirements, we fixed C=7 and λ=0.3, and then determined the optimal $\sigma_{1}$ and $\sigma_{2}$. As can be seen in Figure 4, when $\sigma_{1}$ and $\sigma_{2}$ were within the interval [15,25], the prediction accuracy was higher. Taking $\sigma_{1}$ as an example, we further studied the variation of prediction accuracy with kernel parameters. 

Figure 5 and Table 1 show the prediction accuracy corresponding to different $\sigma_{1}$ values. The highest prediction accuracy was achieved at $\sigma_{1}=20$; therefore, $\sigma_{1}=20$ was chosen in this paper. Using the same method, we obtained the optimal $\sigma_{2}=15$ in this case.

### 5.2. Comparison of the Forecast Results Obtained with Different Inputs 

According to the parameter selection method described in Section 4.3 and Section 5.1, the parameters were set as: C=7, λ=0.3, $\sigma_{1}=20$, $\sigma_{2}=15$.

In order to further illustrate the necessity of using the temperature as an input variable, the prediction results of two kinds of inputs were compared with those of using only historical electric energy as input. The final prediction results are shown in Figure 6. It can be seen from Figure 6 that when only historical electric energy was used as input, a better prediction value could be obtained at the smooth point of the curve, but the prediction at the inflection point was not as good as that of the MMCCSVR prediction model with temperature as input. By comparing the error analysis results of the proposed method with those obtained by using only historical EC as input, it was found that the average MAPE was about 3.29% lower than that of the proposed method. Therefore, the prediction accuracy can be improved to a certain extent by using two kinds of input: EC and temperature.

The relative errors of daily forecasts are shown in Table 2. The characteristics of the shopping mall EC vary in the week. From the data, it appears that the EC is low on Monday and peaks on Saturday and Sunday. Between 4 May and 3 June 2018, the weekends were 5–6 May, 12–13 May, 19–20 May, 26–27 May, and 2–3 June. As shown in Table 2, the peak and valley values of EC were accurately predicted in the first half month, whereas the predicted values were mostly lower than the true values in the second half month. According to the analysis of historical electricity and temperature data, the temperature in May 2018 was slightly higher than that in May of the previous year, but the thunderstorm weather in the first half of the month may have affected customer flow and activities in the shopping mall. The weather in the second half of the month continued to be cloudy, and the EC increased, since the temperature was slightly higher than in previous years.

Table 2 shows the predicted values using MMCCSVR and the traditional SVR from 4 May to 3 June 2018, and the MAPE between each predicted value and the corresponding actual value. As can be seen from Table 2, compared with the SVR model, the MMCCSVR model allowed to reduce the MAPE of almost every predicted value, especially when approaching the turning point of fluctuation. The results show that using MMCC as the loss function of SVR can significantly improve the prediction accuracy.

### 5.3. Comparison of Different Forecasting Methods

In order to test whether the proposed model can effectively improve the prediction accuracy, its prediction results were compared with those of other methods. We set the parameters as follows:

SVR: C=7, γ=0.1; 

Back-propagation algorithm (BP): 10 Hidden layer. 

MMCCSVR: C=7, λ=0.3, $\sigma_{1}=20$, $\sigma_{2}=15$; 

The comparison results are shown in Figure 7. As shown in Figure 7, at the inflection point of the curve, the prediction efficiency of MMCCSVR is obviously better than those of other algorithms, showing the robustness and stability of the proposed method. Moreover, the predicted values changed appropriately during the analysis period, even coinciding with the true values at some individual points.

The percentages of relative error of the different methods are shown in Table 3. Table 3 shows that MMCCSVR has better performance compared with other algorithms, displaying improved prediction accuracy of both global and outlier values.

The relative errors of MMCCSVR for different inputs compared with those of other methods from 4 May to 3 June 2018, are presented in Figure 8. As shown in Figure 8, the relative error of MMCCSVR was evenly distributed on both sides of 0 and had little fluctuation. The relative error of MMCCSVR using historical electricity as input was very large, due to the fact that the temperature changes were not taken into account. BP and traditional SVR algorithms were unable to predict electricity accurately under the influence of random factors.

Table 4 proves the superiority of MMCCSVR over single-input MMCCSVR and the traditional SVR and BP algorithm for four error evaluation indexes; the four error evaluation indexes were obviously reduced, and in particular, the MAPE of this method was about 1.79%, and the R^2^ was about 0.9781. Therefore, compared with the SVR method and the BP neural network algorithm, the proposed method effectively improved the prediction accuracy, and when temperature was added as input, the prediction efficiency was further improved.

Considering the change of EC for a shopping mall in different months, we implemented the method by using EC data for a whole year, using 65% of the data as the training set and 35% as the testing set. We selected the EC data from 1 January to 25 August 2018 as the training sample and forecasted the EC data from 26 August to 31 December 2018, one-day ahead. The forecasted results are shown in Figure 9. As shown in Figure 9, when more data were used to validate the model proposed in this paper, the prediction curve was still close to the actual curve. Only a few curve inflection points presented inevitable errors due to random factors and the impact of emergencies, resulting in larger errors. In addition, at the inflection point of the curve, the prediction ability of MMCCSVR was obviously better than those of other algorithms, showing its robustness and stability. The MMCCSVR achieved good overall prediction, with the predicted values even coinciding with the true values at individual points. 

The relative errors of MMCCSVR for different inputs compared with those of other methods for 26 August to 31 December 2018, are given in Figure 10. As shown in Figure 10, the relative error of MMCCSVR is evenly distributed on both sides of 0 and has little fluctuation. BP and traditional SVR algorithms were unable to predict electricity accurately under the influence of random factors. 

Table 5 proves the superiority of MMCCSVR over the traditional SVR and BP algorithm, as shown by four error evaluation indexes; when the experimental data increased, the four error evaluation indexes were obviously reduced. We found that the MAPE of this method was about 3.86% and R^2^ was about 0.9846. Therefore, compared with the SVR method and BP neural network algorithm, the proposed method has good predictive performance which can effectively improve the prediction accuracy. 

## 6. Conclusions

Around the urgent demand of ECF technology for power sales companies under the wave of power system reform, a novel forecast method for ECF, called MMCCSVR, is proposed, combining the SVR with MMCC. First, the user’s characteristics were analyzed to determine the factors influencing EC, indicating that the historical temperature and historical EC are the main influencing factors. Second, in view of the strong fluctuation of EC for small consumers and the large influence of random factors, the MMCCSVR algorithm with excellent performance under non-Gaussian noise was selected to construct the prediction model, which was applied to a shopping mall in Guangzhou. The following conclusions were reached:(1)Compared with the single-input MMCCSVR prediction model, the single-point prediction accuracy was effectively improved, and the average relative error was reduced.(2)Compared with the traditional SVR and other algorithms, the prediction errors of peak and valley values of EC were improved effectively.(3)The prediction error MAPE of this model was 1.79% and met the assessment criteria of power deviation in the location of the shopping mall and the prediction accuracy requirement of the power sales company.

## Figures and Tables

**Figure 1 entropy-21-00707-f001:**
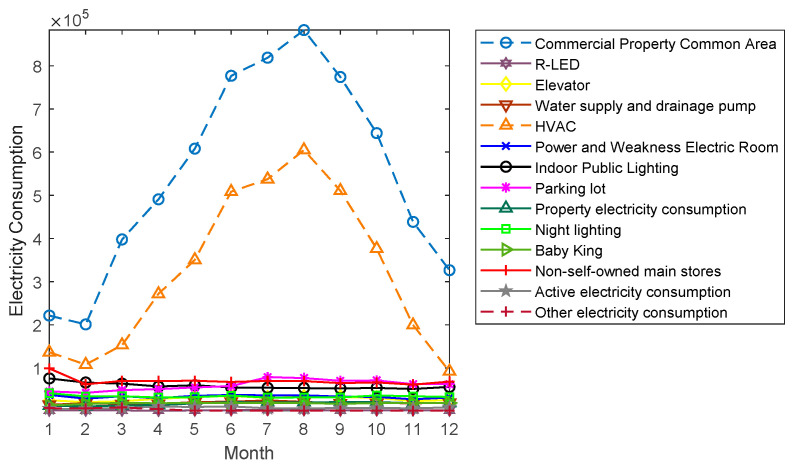
Monthly electricity consumption (EC) of a shopping mall in 2017.

**Figure 2 entropy-21-00707-f002:**
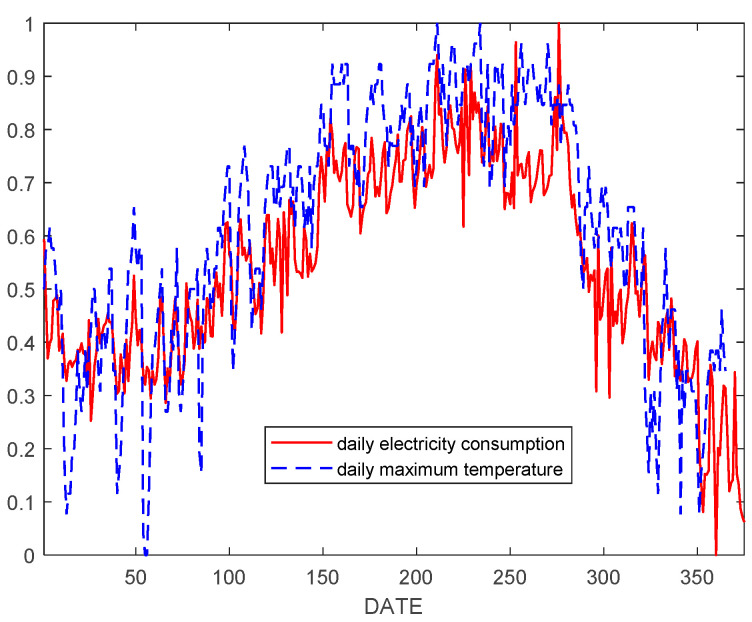
Daily EC of the commercial property common area of the mall and its corresponding daily maximum temperature.

**Figure 3 entropy-21-00707-f003:**
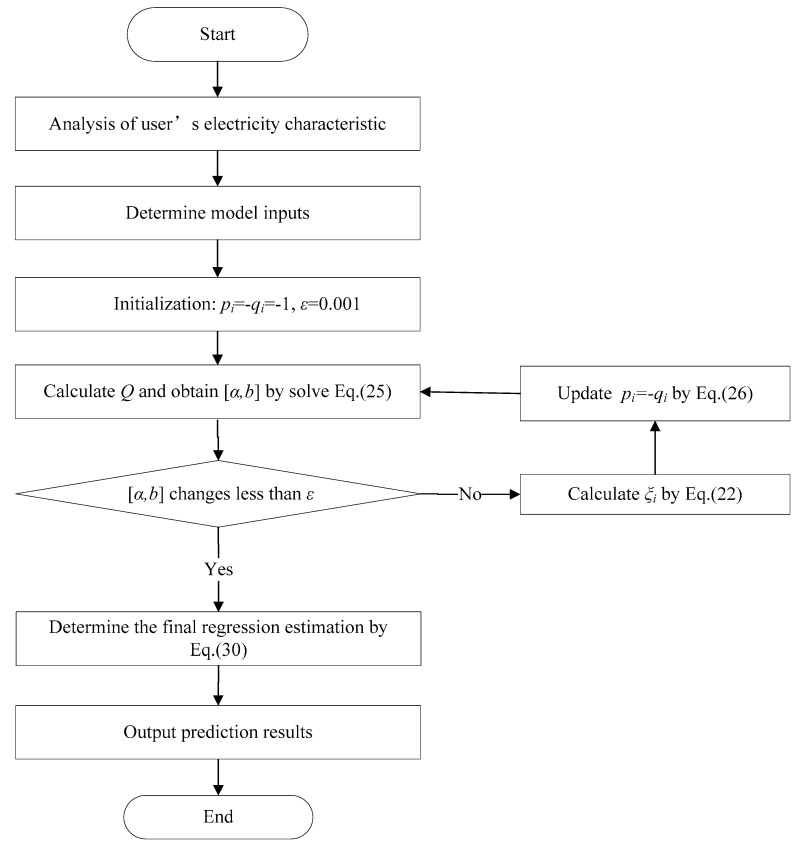
Flow chart of the implementation process.

**Figure 4 entropy-21-00707-f004:**
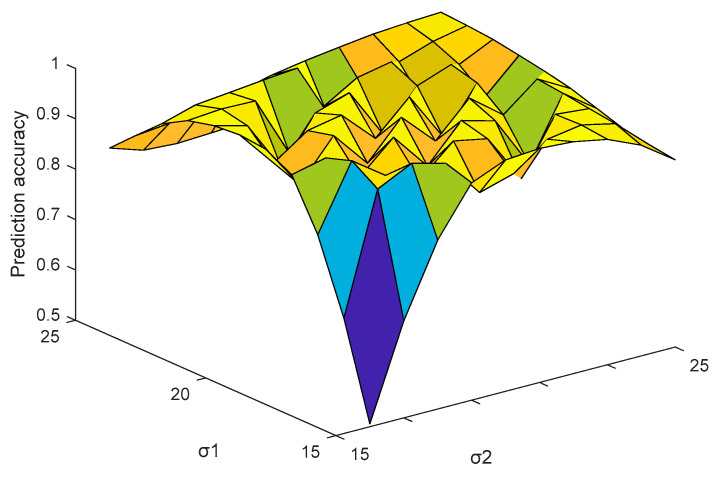
Prediction accuracy varying with the parameters $\sigma_{1}$ and $\sigma_{2}$.

**Figure 5 entropy-21-00707-f005:**
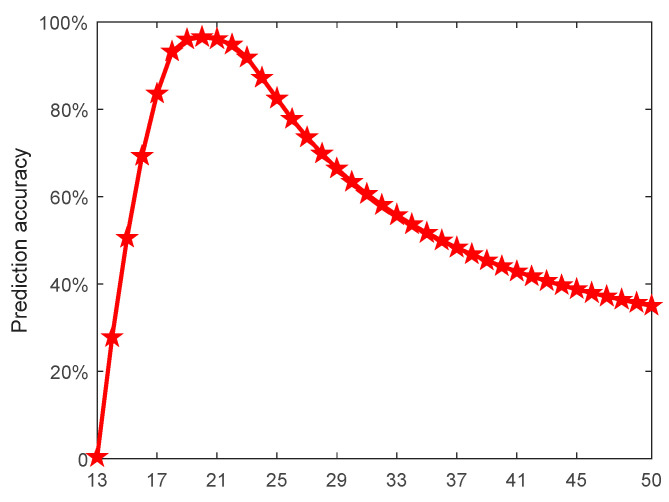
Prediction accuracy varying with the parameter $\sigma_{1}$.

**Figure 6 entropy-21-00707-f006:**
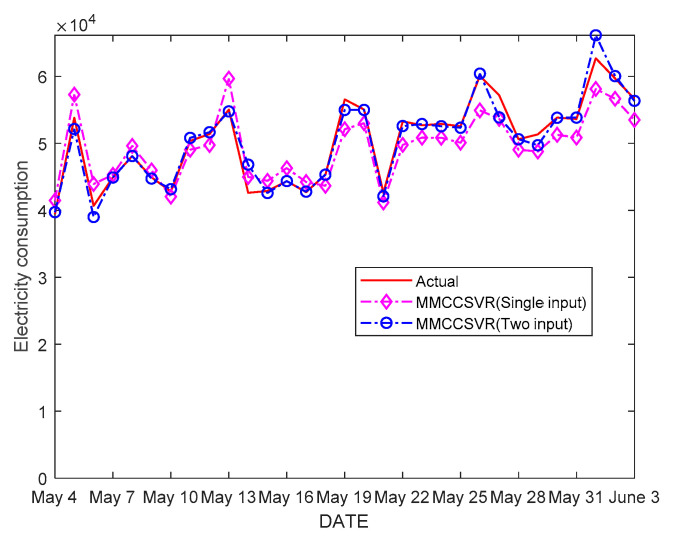
Forecast results of the maximum mixture correntropy criterion support vector regression (MMCCSVR) method for a mall from 4 May to 3 June 2018.

**Figure 7 entropy-21-00707-f007:**
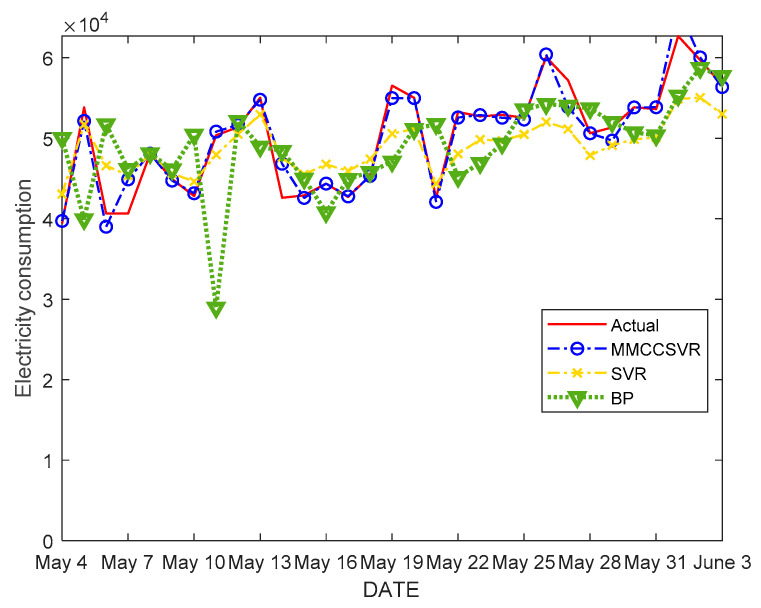
Forecast results of MMCCSVR compared with those of other methods from 4 May to 3 June 2018.

**Figure 8 entropy-21-00707-f008:**
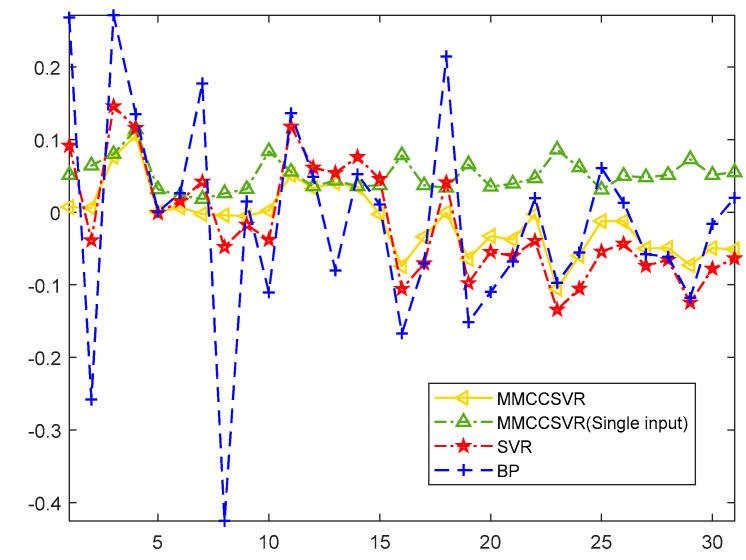
Relative error of MMCCSVR with different inputs compared with other methods from 4 May to 3 June 2018.

**Figure 9 entropy-21-00707-f009:**
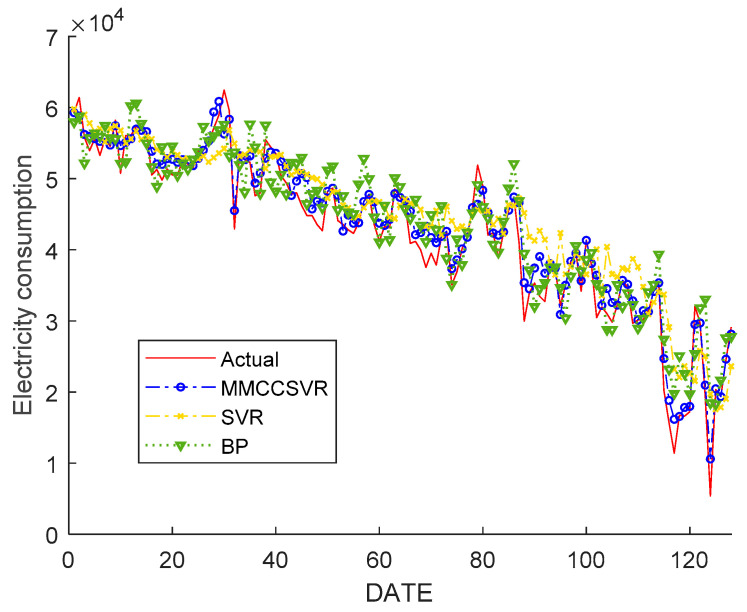
Forecast results of MMCCSVR for a mall from 26 August to 31 December 2018.

**Figure 10 entropy-21-00707-f010:**
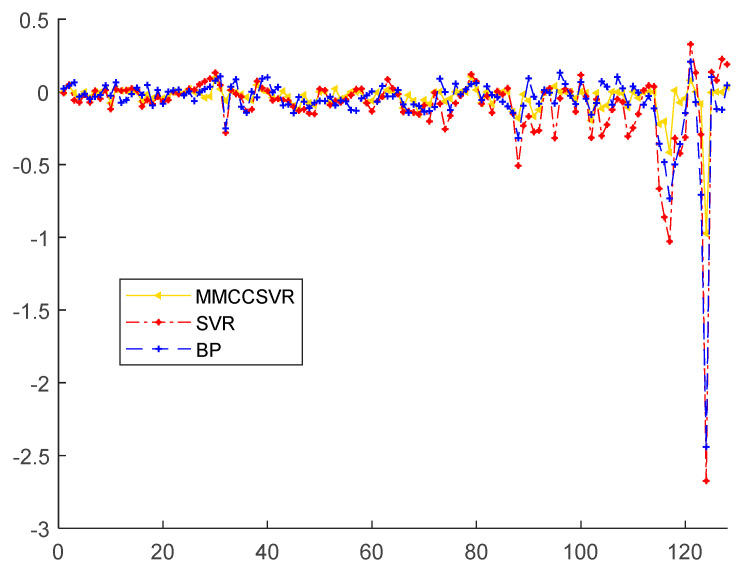
Relative error of MMCCSVR for different inputs compared with those of other methods from 26 August to 31 December 2018.

**Table 1 entropy-21-00707-t001:** Prediction accuracy varying with the parameter $\sigma_{1}$.

$\sigma_{1}$	15	16	17	18	19	20	21	22	23	24	25
**Prediction accuracy (%)**	50.5	69.2	83.6	93.2	96.0	98.2	96.1	94.8	91.8	87.2	82.4

**Table 2 entropy-21-00707-t002:** Forecast results of MMCCSVR for a mall from 4 May to 3 June 2018.

Day	Actual (kW·h)	Single Input (kW·h)	MAPE (%)	Two Input (kW·h)	MAPE (%)
4 May	39,442	41,463	5.12	397,19	0.70
5 May	53,823	41,463	6.43	52,141	−3.13
6 May	40,666	57,283	8.01	38,997	−4.10
7 May	40,666	43,925	11.41	44,889	10.38
8 May	48,099	45,305	3.16	48,100	0.00
9 May	44,929	49,621	2.32	44,735	−0.43
10 May	42,820	45,973	−1.82	43,160	0.79
11 May	50,363	42,039	−2.60	50,820	0.91
12 May	51,381	49,056	−3.18	51,659	0.54
13 May	55,039	59,663	8.40	54,774	−0.48
14 May	42,610	44,984	5.57	46,824	9.89
15 May	42,886	44,408	3.55	42,586	−0.70
16 May	44,358	46,295	4.37	44,364	0.01
17 May	42,699	44,224	3.57	42,777	0.18
18 May	45,329	43,647	−3.71	45,329	0.00
19 May	56,549	52,112	−7.85	54,990	−2.76
20 May	55,039	53,006	−3.69	54,984	−0.10
21 May	42,610	41,170	−3.38	42,090	−1.22
22 May	53,268	49,759	−6.59	52,574	−1.30
23 May	52,707	50,851	−3.52	52,874	0.32
24 May	52,913	50,834	−3.93	52,546	−0.69
25 May	52,547	50,085	−4.69	52,329	−0.41
26 May	60,092	54,925	−8.60	60,408	0.53
27 May	57,188	53,659	−6.17	53,875	−5.79
28 May	50,614	49,047	−3.10	50,614	0.00
29 May	51,341	48,776	−5.00	49,713	−3.17
30 May	53,857	51,274	−4.80	53,825	−0.06
31 May	53,620	50,864	−5.14	53,849	0.43
1 June	62,691	58,118	−7.29	66,143	5.51
2 June	59,699	56,645	−5.12	60,045	0.58
3 June	56,619	53,493	−5.52	56,348	−0.48
MAPE			5.08%	1.79%	

**Table 3 entropy-21-00707-t003:** Percentage of relative error of different methods. BP: Back-propagation.

Day	MMCCSVR	SVR	BP
4 May	0.70	9.17	26.81
5 May	−3.13	−3.88	−25.81
6 May	−4.10	14.59	27.14
7 May	10.38	11.60	13.49
8 May	0.00	−0.12	0.06
9 May	−0.43	1.54	2.64
10 May	0.79	4.21	17.74
11 May	0.91	−4.80	−42.53
12 May	0.54	−1.73	1.49
13 May	−0.48	−3.81	−11.05
14 May	9.89	11.78	13.64
15 May	−0.70	6.12	4.89
16 May	0.01	5.42	−8.01
17 May	0.18	7.59	5.22
18 May	0.00	4.58	1.04
19 May	−2.76	−10.57	−16.71
20 May	−0.10	−7.06	−7.08
21 May	−1.22	4.11	21.46
22 May	−1.30	−9.77	−15.16
23 May	0.32	−5.46	−10.99
24 May	−0.69	−6.04	−6.78
25 May	−0.41	−3.94	1.95
26 May	0.53	−13.44	−9.73
27 May	−5.79	−10.54	−5.53
28 May	0.00	−5.42	6.09
29 May	−3.17	−4.34	1.30
30 May	−0.06	−7.40	−5.79
31 May	0.43	−6.56	−6.13
1 June	5.51	−12.45	−11.81
2 June	0.58	−7.79	−1.61
3 June	−0.48	−6.56	1.96
MAPE	1.79%	6.84%	10.70%

**Table 4 entropy-21-00707-t004:** Comparison of electricity consumption forecasting (ECF) accuracy. MAPE: mean absolute percentage error, MAE: mean absolute error, RMSE: root-mean-square error, R^2^: coefficient of determination.

Method	MAPE	MAE	RMSE	R^2^
MMCCSVR	1.79%	875.8387	1515.228	0.9781
MMCCSVR (Single Input)	5.08%	2582.8387	2836.0348	0.9150
SVR	6.84%	3460.8710	3951.0136	0.9304
BP	10.70%	5220.8065	6957.5602	0.3541

**Table 5 entropy-21-00707-t005:** Forecast results of MMCCSVR for a mall from 26 August to 31 December 2018.

Method	MAPE	MAE	RMSE	R^2^
MMCCSVR	3.86%	1528.2	2289.7	0.9846
SVR	13.78%	3966.4375	6180.0521	0.9173
BP	10.43%	3123.5748	3978.9582	0.9481

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
