# Peer review of "Electricity Consumption Forecasting using Support Vector Regression with the Mixture Maximum Correntropy Criterion"

_entropy, 2019, doi:10.3390/e21070707_

Round 1
Reviewer 1 Report
1. Results: Recommend to be Major revisions
This paper employs mixture maximum correntropy criterion with SVR to improve the accuracy of electricity consumption forecasting. The experiment result shows that proposed ECF method can effectively improve the accuracy of ECF compared with traditional SVR and other forecasting algorithms.
However, readers need more convinced literature reviews as a developed tendency to indicate clearly the state-of-the-art development of SVR hybridizing with other superior algorithm and relevant theoretical background reviews. Thus, authors should not forget the pioneers in the SVR-based fields. I have no ideas why authors do not cite any original innovative papers from the pioneers.
Secondly, authors should provide very details illustrating the procedure how the proposed algorithm is working in the experimental results section, i.e., lacking of the flow chart (Figure 3 is too simple and sometimes naïve) with some essential brief explanation vis-à-vis the text to indicate how the proposed methodology going next.
Finally, authors please conduct some statistical test to verify the significance of the superiority of the results from the proposed approach, i.e., how could authors ensure that their results are superior to others only according to Tables 2 and 4? And authors also require to provide some insight discussion of the results in Tables 2 and 4. Without the significant test, this paper only has minor contribution. Please refer Hong et al. (2019) and Dong et al. (2018).
(1) Hong, W.C.; Li, M.W.; Geng, J.; Zhang, Y. Novel chaotic bat algorithm for forecasting complex motion of floating platforms. Applied Mathematical Modelling 2019, 72, 425-443.
(2) Dong, Y.; Zhang, Z.; Hong, W.C. A hybrid seasonal mechanism with a chaotic cuckoo search algorithm with a support vector regression model for electric load forecasting. Energies 2018, 11(4), 1009.
Author Response
Response to Reviewer 1 Comments
Point 1: This paper employs mixture maximum correntropy criterion with SVR to improve the accuracy of electricity consumption forecasting. The experiment result shows that proposed ECF method can effectively improve the accuracy of ECF compared with traditional SVR and other forecasting algorithms.
Response 1: Thank you for your valuable comments and suggestions, which have helped us to enhance the quality of our paper substantially. We appreciate your expertise and your positive comments.
Point 2: However, readers need more convinced literature reviews as a developed tendency to indicate clearly the state-of-the-art development of SVR hybridizing with other superior algorithm and relevant theoretical background reviews. Thus, authors should not forget the pioneers in the SVR-based fields. I have no ideas why authors do not cite any original innovative papers from the pioneers.
Response 2: Thank you for your valuable comments and suggestions, which have helped us to enhance the quality of our paper substantially. We appreciate your expertise and your positive comments. We have revised the manuscript according to your comments and suggestions and added more literature reviews, and we hope our revised manuscript will meet your approval again.
Point 3: Secondly, authors should provide very details illustrating the procedure how the proposed algorithm is working in the experimental results section, i.e., lacking of the flow chart (Figure 3 is too simple and sometimes naïve) with some essential brief explanation vis-à-vis the text to indicate how the proposed methodology going next.
Response 3: Thanks very much for your suggestions. We have expanded the flow chart according to your suggestion, and explained the specific implementation of MMCCSVR forecasting model in detail in each step. The new flow chart is shown as follows:
Point 4: Finally, authors please conduct some statistical test to verify the significance of the superiority of the results from the proposed approach, i.e., how could authors ensure that their results are superior to others only according to Tables 2 and 4? And authors also require to provide some insight discussion of the results in Tables 2 and 4. Without the significant test, this paper only has minor contribution. Please refer Hong et al. (2019) and Dong et al. (2018).(1) Hong, W.C.; Li, M.W.; Geng, J.; Zhang, Y. Novel chaotic bat algorithm for forecasting complex motion of floating platforms. Applied Mathematical Modelling 2019, 72, 425-443.(2) Dong, Y.; Zhang, Z.; Hong, W.C. A hybrid seasonal mechanism with a chaotic cuckoo search algorithm with a support vector regression model for electric load forecasting. Energies 2018, 11(4), 1009.
Response 4: Thanks very much for your suggestions. We have read and cited the literature you recommended, and in order to verify the significance of the superiority of the results from the proposed approach, we have expanded our description of Table 2 and 4 and added a set of experiment. We implemented the method by using EC data for a whole year, using the 65% samples as training set and the 35% as testing set. We selected the EC data from January 1 to August 25 in 2018 as training samples, and forecasted the EC data from August 26 to December 31, 2018 ahead one-day. The analysis of experimental results has been added in Sec. 5.3.
Reviewer 2 Report
p.p1 {margin: 0.0px 0.0px 0.0px 0.0px; font: 12.0px 'Helvetica Neue'; color: #454545} p.p2 {margin: 0.0px 0.0px 0.0px 0.0px; font: 12.0px 'Helvetica Neue'; color: #454545; min-height: 14.0px} li.li1 {margin: 0.0px 0.0px 0.0px 0.0px; font: 12.0px 'Helvetica Neue'; color: #454545} span.s1 {font: 12.0px '.Hiragino Kaku Gothic Interface'} ul.ul1 {list-style-type: hyphen}This paper aims to introduce an electricity consumption forecasting technique more accurate than the one using the support vector regression (SVR) by replacing SVR with its robust variant against outliers and gaussianity assumption. To do so, this paper attempts to design a robust variant of SVR with use of the mixture maximum correntropy criterion. The resulting robust variant is defined by the optimization problem (12). The authors try to derive an efficient optimization algorithm for (12) with the duality theory in convex optimization and alternating minimization strategy. Section 4 describes how to apply the proposed variant to electricity consumption forecasting, Section 5 confirms performances of the proposed variant in the context of the electricity consumption forecasting in comparison with the one with SVR.
This reviewer recommends the rejection of this paper because the derivation of the proposed alternating minimization strategy contains inaccurate descriptions, which makes me impossible to review this paper.
Proposition 1 seems to be incorrect. The definition of the kernel function kappa is defined in (10) which is free from the parameter lambda. However, (13) in Proposition 1 is involved with the parameter lambda, which is quite strange. I believe that the mixture of two Gaussian (in 11) is utilized in some way for defining the kernel function utilized in (12). Defining the kernel function kappa in (12) is required for reviewing the correctness of this paper.
In Proposition 1, it is unclear that the RHS of (13) is well-defined or not. The existence of the maximum in RHS must be proven rigorously.
The symbols K, e, and Y in (24) are not defined. In addition, the symbol e is also utilized as a variable in (10), which is confusing for readers.
My minor concern are as follows:
In (10), the relation between X,Y and e is ambiguous. Rigorously speaking, (9) does not make sense. The notation \hat{V}(X,Y) implies that \hat is defined by random variables X and Y but the RHS is defined only by their outcomes.
The notation “convex conjugated function” is not standard. Usually, it is referred to as “convex conjugate function”.
Describe clearly whether the proposed algorithm minimization strategy generates a sequence converges to a solution of (12) or not.
In Sec. 5.3, please describe what is BP.
Typo
l.132: K(.) —> K(., .)
l.143: and The —> and the
Author Response
Thank you very much for your valuable comments, here is our response to your comments.

Reviewer 3 Report
Manuscript presents a forecasting method for the electricity consumption based on SVM coupled with the mixture maximum correntropy criterion (MMCC).
The work is found very interesting, relevant to the journal scope, well written and organized with an exhaustive literature review. In the background information as well as the introduction section are adequate to understand the aims and objectives of the study. However, some questions should to be addressed.
- The authors should provide the time horizon of the forecasts. Is it ahead one-day?
- The daily energy consumption data from January 1, 2018 to June 3, 2018 are used to reach the research object. The data from January 1 to May 3 is selected as the training sample and the data from May 4 to June 3, is used as testing sample. Considering the energy daily consumptions for a shopping mall change during months, the authors should implement the method by using daily data for a whole year, using the 65% samples as training set and the 35% as testing set.
- It is noted that the energy demand for the air conditioning depends on the ambient temperature. Therefore, it expects the historical HVAC daily EC series and its corresponding daily maximum temperature series present high Pearson correlation coefficients. In view of this, the authors use the temperature as an input variable for the forecasting method to improve the performance. It is not clear the reason of that.
- Considering the previous the authors could include as input the number of the consumers which is different for each day of a week.
Author Response
Response to Reviewer 3 Comments
Point 1: Manuscript presents a forecasting method for the electricity consumption based on SVM coupled with the mixture maximum correntropy criterion (MMCC).
Response 1: Thank you for your valuable comments and suggestions, which have helped us to enhance the quality of our paper substantially. We appreciate your expertise and your positive comments.
Point 2: The work is found very interesting, relevant to the journal scope, well written and organized with an exhaustive literature review. In the background information as well as the introduction section are adequate to understand the aims and objectives of the study. However, some questions should to be addressed.
Response 2: Thank you for your valuable comments and suggestions, which have helped us to enhance the quality of our paper substantially. We appreciate your expertise and your positive comments.
Point 3: The authors should provide the time horizon of the forecasts. Is it ahead one-day?
Response 3: Thank you for your valuable comments and suggestions. Because of our carelessness, we did not explain the problem. Actually, the time horizon of the forecasts in this paper is ahead one-day. We have added the description in the revised version at the beginning of Sec. 5.
Point 4: The daily energy consumption data from January 1, 2018 to June 3, 2018 are used to reach the research object. The data from January 1 to May 3 is selected as the training sample and the data from May 4 to June 3, is used as testing sample. Considering the energy daily consumptions for a shopping mall change during months, the authors should implement the method by using daily data for a whole year, using the 65% samples as training set and the 35% as testing set.
Response 4: Thanks for your valuable comments and suggestions. For this point, we have implemented the method by using EC data for a whole year, using the 65% samples as training set and the 35% as testing set. We selected the EC data from January 1 to August 25 in 2018 as training samples, and forecasted the EC data from August 26 to December 31, 2018 ahead one-day. The analysis of experimental results has been added in Sec. 5.3.
Point 5: It is noted that the energy demand for the air conditioning depends on the ambient temperature. Therefore, it expects the historical HVAC daily EC series and its corresponding daily maximum temperature series present high Pearson correlation coefficients. In view of this, the authors use the temperature as an input variable for the forecasting method to improve the performance. It is not clear the reason of that.
Response 5: Thank you for your valuable comments and suggestions. For this point, we used the temperature as an input variable for the forecasting method to improve the performance is because of its strong correlation with EC of the shopping mall. Therefore, we assume that the prediction accuracy can be further improved by adding the influence of temperature, and this hypothesis is verified in Sec. 5.
Point 6: Considering the previous the authors could include as input the number of the consumers which is different for each day of a week.
Response 6: Thank you for your valuable comments and suggestions, which have helped us to enhance the quality of our paper substantially. Through the analysis of the experimental results, we did find this law, and mentioned in the paper that the number of the consumers in shopping malls at weekends is bigger than that in working days, so weekend EC is generally higher than that in working days. If the data of the number of the consumers is used as one of the inputs, the prediction accuracy of EC in shopping malls may be further improved. However, due to the limitation of objective factors, it is impossible to collect the number of the consumers in this experiment. We consider that in future work, we can further solve this problem and carry out deeper research.
Round 2
Reviewer 1 Report
Authors have completely addressed all my concerns.
Author Response
Thank you for your approval of our revised draft.
Reviewer 2 Report
Although the revised version has improved the quality of the paper, this reviewer still recommends the rejection of this paper. The derivation of the proposed algorithm seems to be incorrect. Some descriptions remain imprecise.
Equation (15) seems not to hold. I believe that the authors want to find a function \overline{J} satisfying (15) and, to do so, try to apply Proposition 1 to the function g in (12). However, there are two g for each i in (12) and two-times use of Proposition 1 yields two auxiliary parameters. Thus the resulting augmented objective function need to have two auxiliary parameters for each i. Nevertheless, the augmented objective function in (14) has only one auxiliary parameter for each i.
On Point 8, the authors’ response is not sufficient.
———— Excerpt from Response to the first comment
Point 8: Describe clearly whether the proposed algorithm minimization strategy generates a sequence converges to a solution of (12) or not.
Response 8: Thanks so much for your comments. For this point, the optimization problem of equation (12) proposed at the beginning of Sec. 3 has been solved by the proposed algorithm minimization strategy, and we have added the clear description at the end of Sec. 3.
————
Even if the maximization problem (17) comprises the two subproblems that can be solved, the problem (17) is not solved. Hence the sentence in line 208—210 is incorrect. A standard justification that an iterative algorithm solves a problem is to guarantee that the iterative algorithm generates a convergent sequence to a solution of the problem. In this sense, I believe that the proposed algorithm in this paper has no guarantee to solve the maximization problem (17).
Minor comment
l. 183: The notation “convex conjugate function theory” is not standard.
Typo
l. 195,196: [omega, b, p] -> [omega, b, xi]
Author Response
Point 1: Although the revised version has improved the quality of the paper, this reviewer still recommends the rejection of this paper. The derivation of the proposed algorithm seems to be incorrect. Some descriptions remain imprecise.
Response 1: Thank you for your valuable comments, which have helped us to enhance the quality of our paper substantially. We appreciate your expertise and your positive comments.
Point 2: Equation (15) seems not to hold. I believe that the authors want to find a function \overline{J} satisfying (15) and, to do so, try to apply Proposition 1 to the function g in (12). However, there are two g for each i in (12) and two-times use of Proposition 1 yields two auxiliary parameters. Thus the resulting augmented objective function need to have two auxiliary parameters for each i. Nevertheless, the augmented objective function in (14) has only one auxiliary parameter for each i.
Response 2: Thank you so much for your valuable comments, which have helped us to enhance the quality of our paper substantially. Due to our carelessness, we have made some mistakes in the equation’s description which cause you misunderstood what we mean. In lines 195 and 196, there are two errors. They are , not or .To solve this optimization problem, we choose to iteratively optimize (17) by alternatingly optimizing with respect to and while holding the other fix. We can also find the half-quadratic technique is applied to optimize correntropy based cost function in the following references.
[28] He, R.; Hu, B.G.; Zhang W.S. Robust Principal Component Analysis Based on Maximum Correntropy Criterion. IEEE T IMAGE PROCESS 2011, 20(6), 1485-1494.
[29] He, R.; Zheng, W.S.; Tan, T. Half-Quadratic-Based Iterative Minimization for Robust Sparse Representation. IEEE T PATTERN ANAL 2013, 36(2), 261-275.
[30] Wang, Y.; Tang, Y.Y.; Li, L. Correntropy Matching Pursuit With Application to Robust Digit and Face Recognition. IEEE T CYBERNETICS 2016, 47(6), 1354-1366.
We also added these references in the revised version.
Compared with MCC, the MMCC is a weighted combination of two correntropy with different kernel width by using the weight coefficient. Therefore, the establishment of optimization model is the same as MCC optimization in the above references.
We hope the revised version can meet your requirements, and we look forward to receive your comments again.
Point 3: On Point 8, the authors’ response is not sufficient.Even if the maximization problem (17) comprises the two subproblems that can be solved, the problem (17) is not solved. Hence the sentence in line 208—210 is incorrect. A standard justification that an iterative algorithm solves a problem is to guarantee that the iterative algorithm generates a convergent sequence to a solution of the problem. In this sense, I believe that the proposed algorithm in this paper has no guarantee to solve the maximization problem (17).
Response 3: Thank you so much for your valuable comments, which have helped us to enhance the quality of our paper substantially. For this point, we have added the proof of the convergence of proposed method in the revised version. The proof can be shown as follows
Up to now, we have solved both two sub-problems constituting the original optimization problem (17).
For the sake of clarity, we prescribe the proposed MMCCSVR as follows
First, set the parameter values for all samples, tolerance, MMCC parameter and, kernel parameter, regularization parameter, build kernel matrix
Step 1. Calculate diagonal matrix and then solve (25) by Cholesky factorization to obtain
Step 2. If changes less than go to Step 4 else go to Step 3.
Step 3. Calculate error variable by (22) and update by (26) and then go to Step 1.
Step 4. Determine the final regression estimation by (7).
Referring to the convergence analysis method in Ref. [28-30], the convergence of the proposed algorithm is proved as follows
According to step 1 and step 2, we have and .Hence, we can conclude that sequence, is non-decreasing. Based on the property of correntropy, we can verify the objective function is bounded above since and are both bounded above. Due to the above facts and the well-known monotone convergence theorem, the algorithm is guaranteed to converge to its local optimal solution.
Point 4: l. 183: The notation “convex conjugate function theory” is not standard.
Response 4: Thanks so much for your comments. We have corrected the notation “convex conjugate function theory” to “convex conjugated function theory”.
Point 5: l. 195,196: [omega, b, p] -> [omega, b, xi]
Response 5: Thanks so much for your comments. We have made the two mistakes due to our carelessness, which cause you misunderstood what we mean. We have corrected the mistakes in the revised version, they are, not or. We hope the revised version can meet your requirements, and we look forward to receive your comments again.

Reviewer 3 Report
The authors addressed the comment succesfully
Author Response

(The authors gave the same response as above.)
